SciPost Physics

# Topological quantum phase transition of nickelocene on Cu(100)

G. G. Blesio[1], R. Žitko[1,2], L. O. Manuel[3*] and A. A. Aligia[4]

**1** Jožef Stefan Institute, Jamova 39, SI-1000 Ljubljana, Slovenia
**2** Faculty of Mathematics and Physics, University of Ljubljana, Jadranska 19, SI-1000 Ljubljana, Slovenia
**3** Instituto de Física Rosario (CONICET) and Facultad de Ciencias Exactas, Ingeniería y Agrimensura, Universidad Nacional de Rosario, 2000 Rosario, Argentina
**4** Instituto de Nanociencia y Nanotecnología CNEA-CONICET, Centro Atómico Bariloche and Instituto Balseiro, 8400 Bariloche, Argentina
* manuel@ifir-conicet.gov.ar

November 8, 2022

## Abstract

Local quantum phase transitions driven by Kondo correlations have been theoretically proposed in several magnetic nanosystems; however, clear experimental signatures are scant. Modeling a nickelocene molecule on a Cu(100) substrate as a two-orbital Anderson impurity with single-ion anisotropy coupled to two conduction bands, we find that recent scanning tunneling spectra reveal the existence of a topological quantum phase transition from the usual local Fermi liquid with high zero-bias conductance to a *non-Landau* Fermi liquid, characterized by a non-trivial quantized Luttinger integral, with a small conductance. The effects of intermediate valence, finite temperature, and structural relaxation of the molecule position allow us to explain the different observed behaviors.

# 1   Introduction

Quantum phase transitions (QPT) where the Kondo effect is destroyed by competing interactions have been theoretically found in several magnetic nanosystems [1]. They were also invoked to explain the unconventional quantum criticality in some heavy fermion compounds [2]. However, experimental realizations of such QPT [3] are elusive, as they require fine control of the parameter that triggers the transition. Due to their variety and tunability, magnetic single molecules in contact with metal electrodes, probed and manipulated with a scanning tunneling microscope (STM), are distinguished candidate systems. They are being extensively studied because of their novel properties and also their potential use in new spintronic devices [4–7]. In particular, transistors are the most important component of an integrated circuit, as they act as a switch by changing some parameter [7]. We have shown that the the anisotropic two-channel spin-1 Kondo model has a topological QPT changing either the spin-1 anisotropy or the Kondo exchange parameters [8,9] in which the differential conductance $dI/dV$ jumps between zero and the maximum possible value, constituting an ideal switch. In the intermediate-valence regime, the jump is still present although smaller in magnitude [9,10]. In this work we present theoretical evidence that shows that the system of a nickelocene molecule on a Cu(100) substrate is the first known experimental realization of such a molecular switch.

   The double-decker nickelocene (Nc) molecules on the Cu(100) substrates have been experimentally studied in several articles [11–16]. As the STM tip is approached to the molecule, the tunneling regime is followed by the contact regime [13]. While in the latter the differential conductance $dI/dV$ for small bias voltage $V$ displays the characteristic Kondo peak due to screening of the molecular magnetic moment by conduction electrons [17–20], in the tunneling regime $dI/dV$ shows a low-bias dip and finite-$V$ rise characteristic of inelastic spin-flips due to single-ion anisotropy for a spin $S > 1/2$ [21,22]. The different behaviors have been tentatively ascribed to a crossover in the spin of the molecule from $1/2$ in the contact regime to 1 in the tunneling regime based on first-principle calculations [13,16] that, however, neglect dynamical correlations and therefore do not properly treat the Kondo effect. Moreover, the electronic structure does not change much between the two regimes and, as admitted by the authors, the change in the molecular charge is actually insufficient to account for the large change in the magnetization. This calls for an alternative interpretation without such discrepancy.

# 2   Topological quantum phase transition driven by the single-ion anisotropy

The minimal Hamiltonian that captures the many-body physics of this system is an impurity Anderson model in which the dominant configuration has two holes in the Ni 3d shell occupying the nearly degenerate orbitals with $xz$ and $yz$ symmetry, and forming a spin $S = 1$ due to the atomic Hund coupling. These triplet states are split by the single-ion easy plane anisotropy $D$. Both localized orbitals are equally hybridized with conduction electrons with the same symmetry [16]. A recent numerical renormalization group (NRG) study of this class of Hamiltonians [8,10] demonstrated that with the increasing ratio $D/T_K^0$, where $T_K^0$ is the $D = 0$ Kondo temperature, the system undergoes a topological quantum phase transition (TQPT) from a phase with high impurity spectral density $\rho(\omega)$ at zero frequency $\omega$ (proportional to $dI/dV$ for $V = 0$) to a topologically distinct phase with a low $\rho(0)$. Similarly, if the hybridization between localized and conduction electrons $\tilde{V}$ in the Anderson model or the exchange coupling $J$ on the corresponding Kondo model is increased, $T_K^0$ increases, $D/T_K^0$ decreases and the topological transition from low to high $\rho(0)$ is induced.

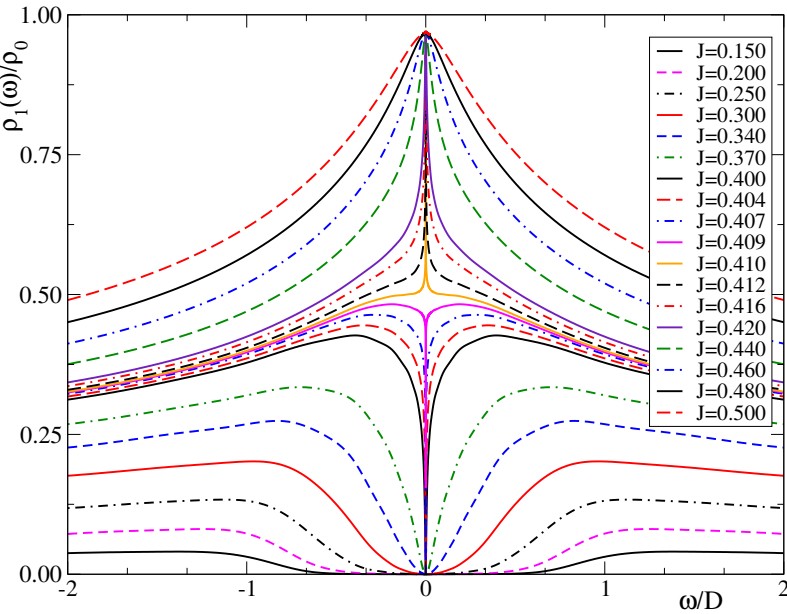

Figure 1: Normalized spectral density of the A2CS1KM as a function of frequency for $D = 0.042W$ and several values of the Kondo coupling $J$, given in units of the half band with $W$ of the conduction channels.

In Fig. 1 we show the evolution of the spectral density for the anisotropic two-channel spin-1 Kondo model (A2CS1KM) [8] (the integer valence limit of the above Anderson model) for fixed $D$ as the Kondo exchange $J$ is increased. For large $J$, $\rho(\omega)$ has the usual shape expected for a Kondo resonance. As $J$ get closer to the critical value $J_c \sim 0.4095$ from above, the shape evolves to a very narrow peak mounted on a much broader one. For $J = J_c$ the narrow peak transforms suddenly into a sharp dip, which broadens with decreasing $J$. For small $J$, $\rho(\omega)$ tends to have two well defined steps at $\omega = \pm D$. A similar behavior has been found in other strongly correlated impurity models [9, 23–25]. The phase with large $D/T_K^0$ is a Fermi liquid characterized by a non-zero Luttinger integral $I_L = \pi/2$ for each orbital and spin [9,10]. Since for a non-interacting system $I_L = 0$, this phase breaks Landau adiabatic hypothesis. Thus for $D/T_K^0 > (D/T_K^0)_c$ the system is a *non-Landau* Fermi liquid (NLFL). Recently, we have shown [9] that this concept can explain in a unified and consistent fashion several experiments in iron phthalocyanine (FePc) on Au(111) [26–28] and we have predicted the possibility of driving that system directly through the TQPT, although such measurement remains to be performed.

The contrasting $dI/dV$ spectra reported in Ref. [13] for Nc on Cu(100) in tunneling and contact regimes coincide with those of the two distinct phases of the model, with the tunneling regime corresponding to the NLFL phase. Moreover, assuming a half band with $W = 1$ eV, the critical anisotropy $D_c \sim 3.8$ meV [10] is of the order of the reported anisotropy $D = 4.2$ meV of the system. Then, it is natural to assume that Nc on Cu(100) is a physical realization of the TQPT. This would be the first experimental realization of a TQPT to a NLFL phase. Previously, we have identified FePc on Au(111), with a clear dip in $dI/dV$ at $V = 0$, as a NLFL [9], using a similar model as the present one with two non-equivalent channels [29]. Nevertheless, an abrupt jump to the conventional Fermi liquid phase has not been reported yet in this system [30].

## 3 Two different behaviors of Nc/Cu(100)

A recent detailed study of Nc/Cu(100) consisting of a careful tip-molecule distance variation [16] revealed more complex behavior that does not quite correspond to that observed in Ref. [13] and shown in Fig. 1. Two different types of evolution of $dI/dV$, probably depending on the exact adsorption geometry, were observed. In case A, the evolution from tunneling to contact regime is discontinuous and hysteretic, with a jump between low and high $dI/dV$, seemingly skipping the regime where a narrow peak/dip for $J \sim J_c$ is expected (as in Fig. 1). In case B, observed experimentally in 1/3 of the cases, the evolution is instead continuous, but it also lacks the narrow peak/dip structures.

To explain these observations, we calculate the phase diagram of the Anderson impurity model [see Eq. (1)] and trace the TQPT as a function of hybridization and energy level. We find that the behavior observed in case A can be explained by a first-order transition in the position of the molecule that avoids parameter values near the TQPT for which the system is less stable or unstable. For this purpose we extend the model by including the effects of the structural relaxation of the molecule, observed in similar systems [27, 31] and expected to play a role also in Nc/Cu(100). For a soft "spring" related to the molecular displacement, we find that the system is unstable near the quantum critical point (QCP) and a first-order TQPT takes place in which the critical point is avoided. The continuous transition experimentally observed in case B is due to two effects: i) the magnitude of the jump in the spectral density at zero frequency $\rho(0)$ decreases as the degree of intermediate valence increases and, more importantly, ii) finite temperatures blur the narrow peaks and dips in $\rho(\omega)$ near the TQPT.

## 4 Topological quantum phase transition at intermediate valence

The Hamiltonian that describes the system can be written in the form [8, 16]

$$
\begin{aligned}
H \;=\; & \sum_{k\tau\sigma} \varepsilon_k c^\dagger_{k\tau\sigma} c_{k\tau\sigma} + \sum_{\tau\sigma} \epsilon\, d^\dagger_{\tau\sigma} d_{\tau\sigma} + \sum_\tau U n_{\tau\uparrow} n_{\tau\downarrow} + \\
& + U' n_{xz} n_{yz} - J_H \vec{S}_{xz} \cdot \vec{S}_{yz} + D S_z^2 + \sum_{k\tau\sigma} \left( \tilde{V} c^\dagger_{k\tau\sigma} d_{\tau\sigma} + \text{H.c.} \right),
\end{aligned}
\tag{1}
$$

where $d^\dagger_{\tau\sigma}$ ($c^\dagger_{k\tau\sigma}$) creates a hole with energy $\epsilon$ ($\varepsilon_k$) in the Ni $d$ orbital $\tau$ (Cu band $\tau$ with momentum $k$), with $\tau = xz, yz$. $n_{\tau\sigma} = d^\dagger_{\tau\sigma} d_{\tau\sigma}$ and $n_\tau = \sum_\sigma n_{\tau\sigma}$. The actual localized orbitals might extend in the Nc molecule beyond the Ni orbitals, but for simplicity we refer to them as the Ni $d$ orbitals. We also assume degenerate conduction bands, hybridizations $\tilde{V}$ independent of energy and channel, and the same energy $\epsilon$ for both orbitals [32]. For small hybridization and when the two-particle configuration dominates, the model (1) reduces to the A2CS1KM [8].

The origin of the anisotropy $D S_z^2$ is the spin-orbit coupling term $H_{\text{SOC}} = \lambda \Sigma \mathbf{l}_i \cdot \mathbf{s}_i$. To provide a qualitative understanding of the effect of $H_{\text{SOC}}$. let us consider the effect of the $z$ components only $H^z_{\text{SOC}} = \lambda \Sigma l^z_i \cdot s^z_i$ on the triplet ground state of the configuration with two holes. This triplet, using the notation $|SS_z\rangle$ for total spin and projection can be written as

$$
\begin{aligned}
|11\rangle \;=\;& d^\dagger_{1\uparrow} d^\dagger_{-1\uparrow} |0\rangle, \quad |1-1\rangle = d^\dagger_{1\downarrow} d^\dagger_{-1\downarrow} |0\rangle, \\
|10\rangle \;=\;& \frac{1}{\sqrt{2}} \left( d^\dagger_{1\uparrow} d^\dagger_{-1\downarrow} + d^\dagger_{1\downarrow} d^\dagger_{-1\uparrow} \right) |0\rangle,
\end{aligned}
\tag{2}
$$

where the operators with orbital projection $l_z = \pm 1$ are $d^{\dagger}_{\pm 1 \sigma} = (\mp d^{\dagger}_{xz\sigma} - i d^{\dagger}_{yz\sigma})/\sqrt{2}$. Then

$$H^z_{\text{SOC}}|1 \pm 1\rangle = 0, \ H^z_{\text{SOC}}|10\rangle = \lambda|00\rangle, \tag{3}$$

where the singlet $|00\rangle$ has the same form as $|10\rangle$ but with a minus sign instead of plus in Eq. (2), and it is higher in energy by $J_H$. Furthermore, $H^z_{\text{SOC}}|00\rangle = \lambda|10\rangle$. Thus, to second order in perturbation theory in $H^z_{\text{SOC}}$, the energy of the states $|1 \pm 1\rangle$ remains the same, but the energy of $|10\rangle$ is lowered by $D \simeq \lambda^2/J_H$. Taking $\lambda \sim 80$ meV and $J_H = 1.5$ eV, $D \sim 4$ meV in good agreement with the experimental value 4.2 meV [16].

The calculations were done with the NRG Ljubljana [33,34] implementation of the numerical renormalization group method [35,36]. The Hamiltonian was implemented with conserved total charge as well as the conservation of the $z$-component of the total spin, i.e., $U(1) \times U(1)$ symmetry. We kept up to 10000 multiplets (or up to cutoff 10 in energy units) in the truncation with the discretization parameter $\Lambda = 4$ and averaging over $N_z = 4$ different discretization meshes. The spectral functions were computed using the complete Fock space algorithm [37], and the resolution for the Anderson model was improved using the "self-energy trick" [38].

For the discussion of topological properties, we remind the reader that the impurity spectral function *per* orbital and spin, at the Fermi level ($\omega = 0$) at zero temperature, is related to the quasiparticle scattering phase shift $\delta_{\tau\sigma}$ by [39–45]

$$\rho_{\tau\sigma}(0) = -\frac{1}{\pi}\text{Im}G^d_{\tau\sigma}(0) = \frac{1}{\pi\Gamma}\sin^2\delta_{\tau\sigma}, \tag{4}$$

where $G^d_{\tau\sigma}(\omega) = \langle\langle d_{\tau\sigma}; d^{\dagger}_{\tau\sigma}\rangle\rangle$ is the impurity Green's function and $\Gamma = \pi \sum_k |\tilde{V}|^2 \delta(\omega - \varepsilon_k)$ is the hybridization strength, assumed independent of energy. In turn, the phase shift is related to the number of displaced electrons by the impurity for each channel and spin which, in the case of wide flat conduction density of states, coincides with the expectation value of the Ni occupancy for the corresponding spin and orbital. Therefore, in this limit, the generalized Friedel sum rule reads [9,10]

$$\delta_{\tau\sigma} = \pi\langle n_{\tau\sigma}\rangle + I_{\tau\sigma}, \ I_{\tau\sigma} = \text{Im}\int_{-\infty}^{0} d\omega \, G^d_{\tau\sigma}(\omega)\frac{\partial\Sigma^d_{\tau\sigma}(\omega)}{\partial\omega}, \tag{5}$$

where $\Sigma^d_{\tau\sigma}(\omega)$ is the impurity self energy. In our case, by symmetry, the four integrals are equal, $I_{\tau\sigma} = I_L$. Until recently, $I_L = 0$ was generally assumed as a hallmark of a Fermi liquid, but several local Fermi liquids were found in which $I_L = \pm\pi/2$ [8–10,24,25].

Since $I_L = (\pi/2)\theta(D - D_c)$ [8–10], Eqs. (4) and (5) imply that for a total Ni occupancy of $\langle n\rangle = 4\langle n_{\tau\sigma}\rangle = 2$, the spectral densities $\rho_{\tau\sigma}(0)$ jump from $\rho_0 = 1/\pi\Gamma$ for $D < D_c$ to 0 for $D > D_c$. For fixed $D$ the TQPT can be obtained by decreasing $\Gamma$ or the exchange constant $J$, as displayed in Fig. 1. For other occupancies, the jump in the spectral densities at the TQPT is reduced as $\Delta\rho_{\tau\sigma}(0) = -\frac{1}{\pi\Gamma}\cos(\pi\langle n\rangle/2)$.

Clearly this relation predicts that the jump disappears as the Ni occupancy is reduced to one particle. More fundamentally, the formation of the $S = 1$ state requires a strong contribution of the two-particle configuration in the ground-state wave function. This naturally leads to the question of the very existence of the TQPT away from half-filling. To elucidate this, we have calculated the position of the TQPT in the $\epsilon, \Gamma$ plane. We take similar parameters for the model as those reported previously [16] with the half band width $W = 1$ eV as the unit of energy and a flat density of states for the conduction electrons. The result is displayed in Fig. 2. For $\epsilon = -(U + U')/2 = -4.25$, $\langle n\rangle = 2$ and $\Gamma_c$ reaches its maximum value $\sim 350$ meV. As $\epsilon$ increases one expects that the Kondo temperature increases for constant $\Gamma$, and then to keep the same ratio $D/T_K \sim 2.6$ at the TQPT [10], $\Gamma_c$ should decrease. This decrease is more abrupt near $\epsilon \sim -2.5$ until for $\epsilon = \epsilon_0 = -2.375$, $\Gamma_c \to 0$ at the TQPT. For larger $\epsilon$, the system is always in the topologically trivial ($I_L = 0$) phase.

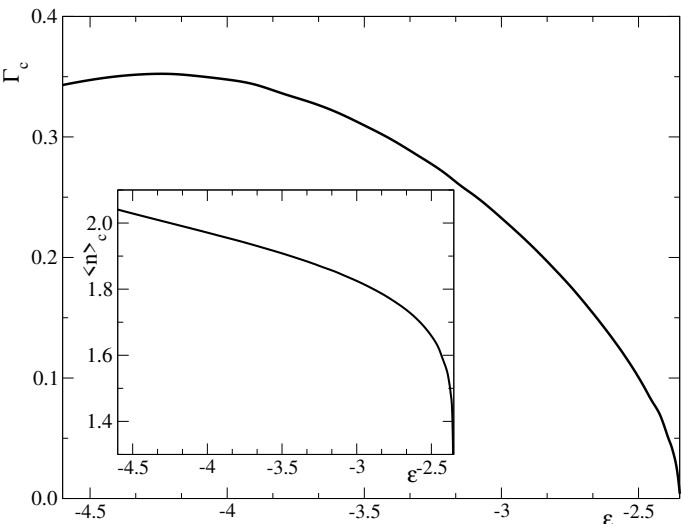

Figure 2: Phase diagram of the Anderson model in the $\epsilon, \Gamma$ plane for $U = 3.5$, $U' = 2.5$ and $J_H = 0.5$. Below (above) the curve the phase is a NLFL (usual Fermi liquid). Inset: occupancy of the Ni at the TQPT.

Note that the end point of the TQPT $(\epsilon, \Gamma) = (\epsilon_0, 0)$ is the point of degeneracy of the ground state configurations of the isolated molecule ($\Gamma = 0$) between the two-particle state with $S_z = 0$ [$|10\rangle$ given by Eq. (2) with energy $2\epsilon + U' - J_H/4$] and the four one-particle states ($d_{\pm 1\sigma}^\dagger |0\rangle$ with energy $\epsilon$). A TQPT at this point might be expected on general physical grounds. If the one-particle configuration is that of lowest energy, the splitting of the orbitals is physically relevant and cannot be neglected for a realistic description (as for FePc [29]). Neglecting the excited doublet, the model mixes the doublet of the states $d_{1\uparrow}^\dagger |0\rangle$, and $d_{-1\downarrow}^\dagger |0\rangle$ with the triplet given by Eqs. (2). For $D = 0$, the model has been solved exactly [46] and the ground state is a triplet. For arbitrary $D$ and small $\Gamma$, a Schrieffer-Wolf transformation like that performed in Ref. [46] leads to the following Kondo interaction

$$H_I = \left( \frac{-2\tilde{V}^2}{\Delta E + D} + \frac{\tilde{V}^2}{\Delta E} \right) S_d^z S_c^z - \frac{\tilde{V}^2}{\Delta E} \left( S_d^x S_c^x + S_d^y S_c^y \right), \tag{6}$$

where $\vec{S}_d$ and $\vec{S}_c$ are the total spin of the localized and conduction electrons respectively, and $\Delta E$ is the excitation energy from the one-particle GS configuration to the $S_z = 0$ triplet excited state. According to poor man's scaling [47], for $D > 0$, the ground state is a singlet and corresponds to an ordinary Fermi liquid as we found. If instead, the two-particle configuration dominates, the model is equivalent to the A2CS1KM [8] with exchange $J \to 0$ which is a NLFL.

In the inset of Fig. 2 we show the Ni occupancy $\langle n \rangle$ along the TQPT line. It lies always above $\sim 1.5$. For $\langle n \rangle = 1.5$, the jump in the spectral density is reduced by a factor 0.71. Therefore, while the TQPT is present with some degree of intermediate valence, it seems that the two-particle triplet Ni configuration should be the dominant one for the TQPT to be observed.

We now discuss the evolution of $dI/dV$ for type B [16], where a jump the spectral density at the TQPT is not observed, but there is a rather continuous evolution as the STM tip is pressed against the surface, increasing $\Gamma$. As discussed above, the jump can be reduced up to 30% by the effects of intermediate valence. Such interpretation is supported by the measured conductances that saturate well below the unitary limit (see Fig. 4 in Ref. [16]). Nevertheless, an even more important effect is that due to the finite temperature of the experiment ($T = 4.5$ K): the narrow dip/peak near the TQPT (see Fig. 1) is blurred by temperature and, furthermore, at finite temperature $dI/dV$ is not proportional to $\rho_{\tau\sigma}(\omega)$ but to its convolution with

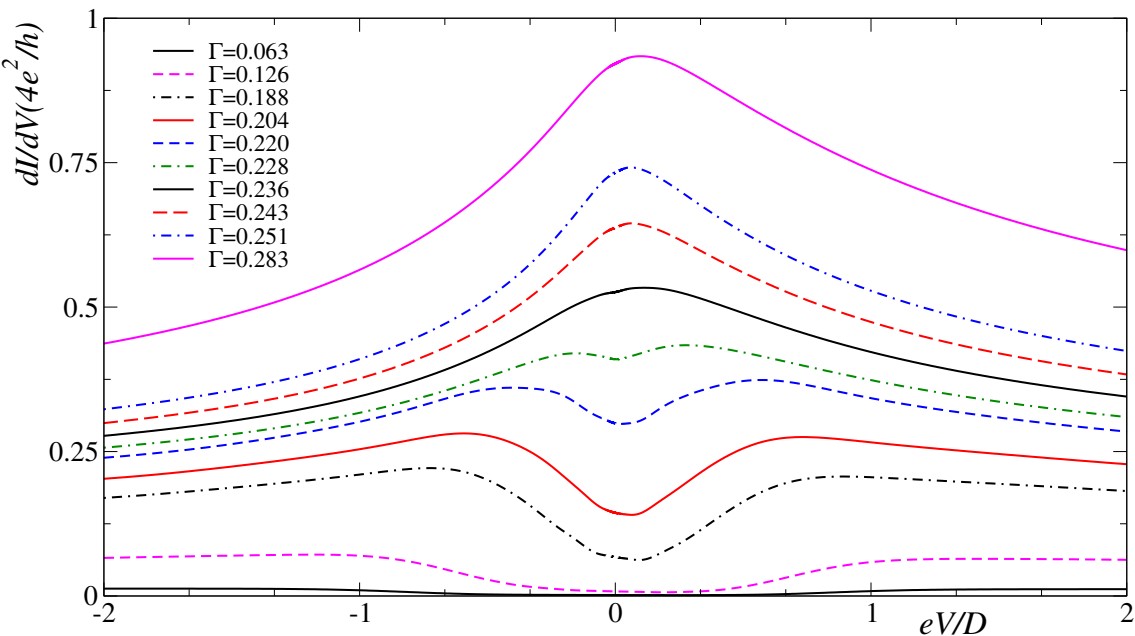

Figure 3: Differential conductance as a function of voltage for different values of $\Gamma$. Other parameters are $U = 3.5$, $U' = 2.5$, $J_H = 0.5$, $\epsilon = -3.0$, $T = 0.0005$.

the derivative of the Fermi function (see for example Ref. [9]) that further broadens sharp features near the TQPT.

In Fig. 3 we show the evolution of the differential conductance as $\Gamma$ is increased at a finite temperature. For small $\Gamma$, $dI/dV$ has a dip mounted on a broader peak. As $\Gamma$ increases, the dip narrows, but in contrast to the case of zero temperature, the minimum of the dip increases and a very sharp dip like that of Fig. 1 is absent. The effect of temperature on the dip is similar to that shown in Fig. 3 of Ref. [9]. For larger $\Gamma$ the dip disappears and the magnitude of $dI/dV$ near zero voltage increases. A sharp peak like that of Fig. 1 for $J$ slightly above the transition value $J_c$ is also absent, mainly as an effect of finite temperature. Therefore, the main features of Fig. 5 (b) of Ref. [16] are reproduced. Our curves seem somewhat more asymmetric than the experimental ones. This can be corrected by decreasing $\epsilon$.

## 5 Structure relaxation of the nickelocene molecule and first-order transition

To explain the evolution of $dI/dV$ for the experimental situation of type A (see Section 3 for the explanation of types A and B), we need to extend the model to permit the structural relaxation of the molecule. Clearly $\Gamma$ increases as the STM tip approaches to the Cu(100) surface, and (as in other systems [27, 31]) the Nc molecule accommodates itself to the tip position, mainly tilting its axis relative to the surface. This shift in the position of the molecule, which we denote as $\eta$, also affects $\Gamma$. On general physical grounds one expects, to leading order in $\eta$,

$$\Gamma = \Gamma_0 + a\eta, \tag{7}$$

where $\Gamma_0$, which depends on the position of the STM tip, is the value of $\Gamma$ for $\eta = 0$ and $a$ is a constant. The on-site energy $\epsilon$ might depend slightly on $\eta$ but this does not affect the main argument below. The total Hamiltonian $H_t$ should also include the elastic energy due to the

displacement $\eta$,

$$H_t = H + \frac{1}{2}K\eta^2, \tag{8}$$

where $K > 0$ is another constant. The optimum value of $\eta$ for each $\Gamma_0$ is obtained by minimizing the total energy $E_t = \langle H_t \rangle = E(\Gamma) + b\eta^2$, where $E = \langle H \rangle$. Differentiating this equation one has

$$\frac{dE_t}{d\eta} = \frac{\partial E}{\partial \Gamma}a + K\eta, \quad \frac{d^2E_t}{d\eta^2} = \frac{\partial^2 E}{\partial \Gamma^2}a^2 + K. \tag{9}$$

In order to have a locally stable minimum at the value $\eta = \eta_1$ for which $dE_t/d\eta = 0$, one needs that $d^2E_t/d\eta^2 > 0$ for $\eta = \eta_1$. If, however, for some value $\Gamma = \Gamma_u$, $\partial^2 E/\partial \Gamma^2 < -K/a^2$, the position $\eta = \eta_1$ of the molecule is unstable and, as $\Gamma_0$ is varied, the system has a first-order transition between two states, one with $\Gamma < \Gamma_u$ and another one with $\Gamma > \Gamma_u$, avoiding the value $\Gamma = \Gamma_u$.

To analyze this possibility in our system, we have calculated the ground state energy $E(\Gamma)$ and numerically differentiated it, see Fig. 4. Remarkably, $\partial^2 E/\partial \Gamma^2$ is negative and large in magnitude at the value $\Gamma = \Gamma_c$ that corresponds to the TQPT. Therefore, one expects that for a soft enough spring constant ($K < a^2|\partial^2 E/\partial \Gamma^2|$), as the STM tip is pressed into the molecule, there is a first-order transition from the NLFL phase to the usual Fermi liquid phase, avoiding the values $\Gamma \sim \Gamma_c$. This agrees with the experimental observations reported in Ref. [13] and Fig. 5 (a) of Ref. [16] for systems of type A. For the cases of type B, it is likely that relaxation of the molecule is inhibited by a larger spring constant $K$. In the contact regime, the Kondo resonance is much broader for the type B situation than for the type A, suggesting an upright molecular configuration in case B and a tilted one in case A [16].

# 6 Conclusion

We have uncovered a topological quantum phase transition (TQPT) in published tunneling spectra of nickelocene molecules on Cu(100). The TQPT is controlled by the distance between the STM tip and the surface. We have modeled the system as a spin-1 two-channel Anderson model with easy-plane magnetic anisotropy, which has a TQPT between a non-trivial topological *non-Landau* Fermi liquid phase in the tunneling regime to a topologically trivial Fermi liquid phase in the contact regime. The transition extends to the zero-hybridization limit at the degeneracy point between the two-particle configuration and the one-/three-particle configuration. We find that the different behaviors of the differential conductance $dI/dV$ as a function of voltage can be explained in terms of this TQPT. The transition is characterized by a jump in $dI/dV$ at $V = 0$ at zero temperature. However, in most experimental observations (case A) there is a hysteretic jump in $dI/dV$ also at finite voltage which can be explained by the structural relaxation of the molecule. In the absence of this relaxation and at finite temperature, the change in $dI/dV$ is continuous as the tip of the STM is approached to the molecule (case B). To our knowledge, this is the first experimental realization of the TQPT and might have applications in molecular electronics. We hope that our work will stimulate further research and help the interpretation of ongoing experiments in similar systems.

# Acknowledgements

RŽ acknowledges the support of the Slovenian Research Agency (ARRS) under P1-0044 and P1-0416. GGB is also supported by the Slovenian Research Agency (ARRS) under P1-0044

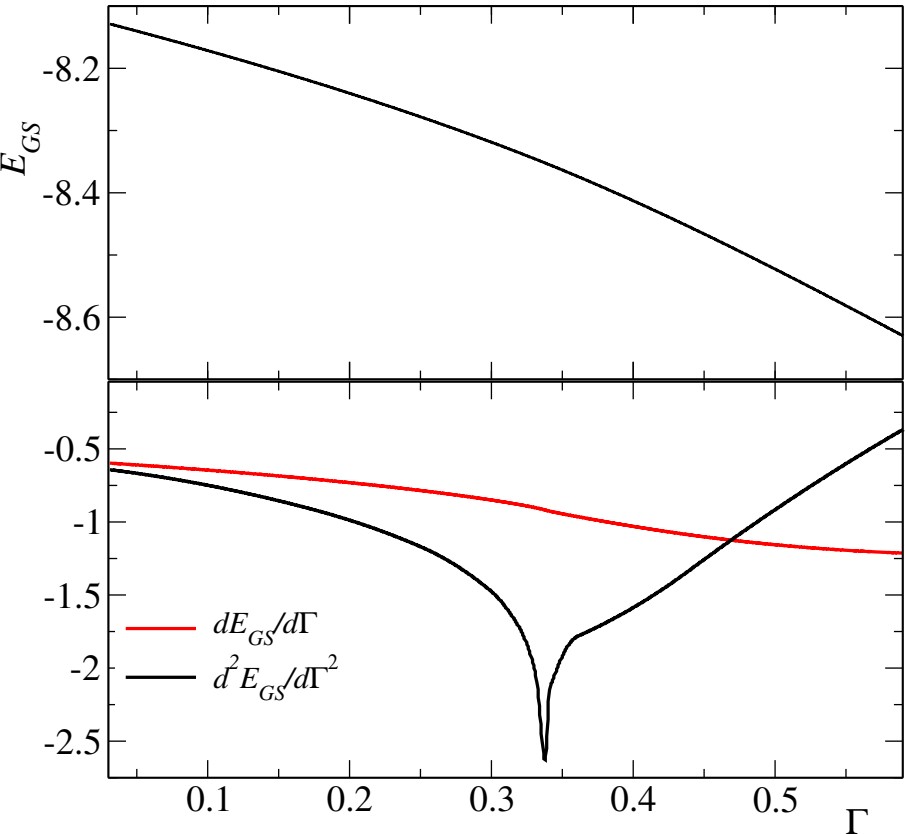

Figure 4: Ground-state energy and its first and second derivatives as a function of $\Gamma$ for $U = 3.5$, $U' = 2.5$ and $J_H = 0.5$, and $\epsilon = -3.8$.

and J1-2458. GGB and LOM are supported by PIP No. 3220 of CONICET, Argentina. AAA is supported by PICT 2017-2726 and PICT 2018-01546 of the ANPCyT, Argentina.

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
