# Peer review of "Topological quantum phase transition of nickelocene on Cu(100)"

_SciPost Physics, doi:SciPost Phys. 14, 042 (2023)_

## Round 1 · Referee Report · Anonymous (Referee 1) · 2022-10-12

Report

The authors study local quantum phase transitions in a two-orbital Anderson model describing nickelocene on Cu(100). STM experiments have shown that the tunneling spectrum strongly depends on the distance between tip and molecule (on the hybridization strength between tip and molecule). In particular, the molecule undergoes a quantum phase transition between a phase with a Kondo peak at the Fermi energy in the spectrum to a phase where the spectral weight around the Fermi energy is strongly suppressed.
While there have been calculations using the anisotropic two-channel spin- 1 Kondo model, recent STM experiments show spectra that cannot be directly explained by existing calculations. While the experiments show a quantum phase transition, a small Kondo peak, observed close to the quantum phase transition in the spin model, is absent in the experiment. Furthermore, in some experiments, a first-order transition has been observed.

In the current paper, the authors use a two-orbital Anderson model to describe these STM experiments consistently. The main findings are
-By including a local energy, the spectra become consistent with the experiment.
-The quantum phase transition is a topological quantum phase transition. For one phase, the Luttinger integral does not vanish.
-This topological quantum phase transition persists for finite doping until the particle number expectation value becomes one.
-Finally, by considering a tilting of the molecule, the phase transition becomes a first-order transition for certain parameters of the spring constant.

This paper gives a consistent explanation of the STM spectra. Furthermore, it demonstrates that the observed quantum phase transition is a topological quantum phase transition. I believe this manuscript solves this problem and has the potential for follow-up work.

I have the following comments and questions:
This is a topological quantum phase transition, where the Luttinger integral changes its value.
Besides the change of this integral, is there any other consequence for the properties of the system originating from the "topology"?
I think it would help to see the Green's function and the self-energy in both phases. Can the authors include a figure so that the reader sees how the self-energy changes at this phase transition?
Below Fig. 1, the authors say that the phase with large D is a Fermi liquid phase where the Luttinger integral does not vanish. However, in Fig. 1, only J is varied. Does the topology change in this figure?

The small Kondo peak that is present in Fig. 1 (which was not observed in the experiment) and absent in Fig. 3, how do these two tunneling spectra connect to each other? At epsilon=-4 (approximately half-filling), the peak is present, but at epsilon=-3, it is absent. How does this change occur?

Due to the doping, the tunneling spectra in Fig. 3 are asymmetric. However, the STM experiments (PHYSICAL REVIEW B 101, 075414) show symmetric spectra. Is it possible to reproduce the same evolution of tunneling spectra (consistent with the experiment) that are symmetric?

Above equation 4, the authors explain the ground state impurity configuration necessary for the topological quantum phase transition.
"Neglecting the excited doublet, the model mixes a doublet and a triplet. ...". Without further explanation, it is very difficult to understand what doublet or triplet the authors are speaking about. I suggest extending this part.

In summary, this manuscript is a consistent explanation of recent experiments that demonstrate the existence of a topological quantum phase transition. I will support the publication of this manuscript if the authors appropriately answer my questions.
  • validity: -
  • significance: -
  • originality: -
  • clarity: -
  • formatting: -
  • grammar: -

Author:  Luis Manuel  on 2022-11-08  [id 2992]

(in reply to Report 1 on 2022-10-12)

The referee writes:
This is a topological quantum phase transition, where the Luttinger integral changes its value.
Besides the change of this integral, is there any other consequence for the properties of the system originating from the "topology"?

Our response:
This is an interesting question. Unfortunately, we do not have a good answer. For large D, there is a zero of the Green’s function at zero energy. Gurarie and coworkers have shown that the presence of such zeros modifies the bulk-boundary correspondence (Eq. 11 of Manmana et al., Phys. Rev. B 86, 205119 (2012)). Perhaps this can have an effect for the boundary of the extension of our model to a periodic system, which might be solved with the dynamical mean-field theory.

The referee writes:
I think it would help to see the Green's function and the self-energy in both phases. Can the authors include a figure so that the reader sees how the self-energy changes at this phase transition?

Our response:
The calculation of the self-energy is technically complicated, and particularly difficult near a divergence. Some of the present authors showed the real and imaginary part of the self-energy in Fig. S4 of the Supplemental Material of Ref. 10, but the results are not quite reliable. In any case, one can see from those figures that the self energy is featureless for small D (ordinary Fermi liquid phase) and approaches a singular behavior in the non-Landau phase, particularly near the transition. More accurate results for the imaginary part were presented in Fig. 2 of the Supplemental Material of Ref. 9 in presence of a small magnetic field which shifts the pole from zero energy. These results, as well as the analytical treatment of that Supplemental Material, are consistent with the presence of a pole in the self-energy at the Fermi level in the non-Landau phase. Unfortunately, to include such figures with the necessary accuracy would require several months, and is a rather technical point that does not affect the comparison with experiment.

The referee writes:
Below Fig. 1, the authors say that the phase with large D is a Fermi liquid phase where the Luttinger integral does not vanish. However, in Fig. 1, only J is varied. Does the topology change in this figure?

Our response:
Yes. In the new version, we have added a sentence on page 2, clarifying that with increasing J, the topological transition takes place in the opposite sense as with increasing D (the Luttinger integral is not trivial for large D or small J).

The referee writes:
The small Kondo peak that is present in Fig. 1 (which was not observed in the experiment) and absent in Fig. 3, how do these two tunneling spectra connect to each other? At $\epsilon= -4$ (approximately half-filling), the peak is present, but at $\epsilon=-3$, it is absent. How does this change occur?

Our response:
As explained in more detail at the end of Section 4, this is mainly an effect of finite temperature, which strongly reduces the magnitude of the dip (peak) for $\Gamma$ or J slightly below (above) the critical values. The change in occupancy from half filling, also has an effect of decreasing the magnitude of the peak and increasing the minimum value of the conductance, according to the generalized Friedel sum rule, Eqs. (2) and (3). However, the main effect is due to finite temperature.

The referee writes:
Due to the doping, the tunneling spectra in Fig. 3 are asymmetric. However, the STM experiments (PHYSICAL REVIEW B 101, 075414) show symmetric spectra. Is it possible to reproduce the same evolution of tunneling spectra (consistent with the experiment) that are symmetric?

Our response:
Experimentally, the spectra are also asymmetric, although to a lesser extent. For $\epsilon$ closer to half-filling, the calculated spectra are more symmetric. A short comment on this point is added in the paper. We did not attempt to obtain quantitative agreement. Instead, as explained in the text, we took the main parameters as those used in the paper by Mohr et al. without fine tuning. Our main goal was to obtain a continuous evolution of the curves as in experiment, instead of the jump as in Fig. 1.

The referee writes:
Above equation 4, the authors explain the ground state impurity configuration necessary for the topological quantum phase transition.
"Neglecting the excited doublet, the model mixes a doublet and a triplet. ...". Without further explanation, it is very difficult to understand what doublet or triplet the authors are speaking about. I suggest extending this part.

Our response:
In the revised version of the paper, we write explicitly the doublet and the triplet states.

---

## Round 1 · Referee Report · Anonymous (Referee 2) · 2022-10-17

Strengths

1 - Demonstrates an experimental realization of a novel unconventional topological phase transitions driven by Kondo correlations.
2 - Offers a relatively simple explanation in terms of a 2 channel Anderson impurity model that accounts for all the experimental observations.

Weaknesses

1 - None.

Report

This paper explores a topological phase transition induced by Kondo correlations in a spin S=1 two-channel Anderson model. The two orbital model studied here describes the d_xz and d_yz of a transition metal (in this case Ni in a nickelocene molecule) with intra and inter orbital interactions and Hund exchange plus the addition of a single ion anisotropy D. By means of exhaustive NRG calculations the authors are able to determine the phase boundary of the so called "non-Landau Fermi liquid phase" and show that it survives beyond half-filling. Moreover, they calculate the experimental signatures of this transition through the dI/dV curves as a function of the distance between and STM tip and the molecule. By taking into account the effects of the elastic energy as the tip pushes the molecule into the surface they show how the transition becomes first order if accompannied by a distortion. This elegant description gives a compelling explanation for the observed experimental results. I recommend this paper for publication after some minor observations are addressed.

Requested changes

1- The authors justify the model by observing that the two d orbitals couple to channels with different symmetry. They keep only two orbitals because they are the ones where the holes reside forming an affective spin S=1 and are basically constraining the five orbital Kanamori-Anderson model to this relevant manifold. For completeness, it would be useful for the readers if the authors can also justify the origin of the anisotropy D, which is not completely obvious to me and will probably just require a single sentence.

2- The statement explaining the degeneracy between the two and one particle states for gamma=0 can be confusing. There are more than one two particle states with Sz=0: |20>;|02>;|up dn>;|dn up> (before symmetrization). The authors would help the reader by enumerating or explicitly pointing out what states they refer to and justifying why the others are discarded.

3- I suspect there is a typo in Eq. 4; the second term is probably S+S- + hc.

4- In the first sentence of section 5 "the experimental situation of type A" is not defined, what do the authors mean? (type B is later mentioned, but never described). If this requires the addition of a figure, I strongly recomment its addition for clarity.

  • validity: top
  • significance: top
  • originality: top
  • clarity: high
  • formatting: perfect
  • grammar: excellent

Author:  Luis Manuel  on 2022-11-08  [id 2991]

(in reply to Report 2 on 2022-10-17)

The referee writes: 1- The authors justify the model by observing that the two d orbitals couple to channels with different symmetry. They keep only two orbitals because they are the ones where the holes reside forming an affective spin S=1 and are basically constraining the five orbital Kanamori-Anderson model to this relevant manifold. For completeness, it would be useful for the readers if the authors can also justify the origin of the anisotropy D, which is not completely obvious to me and will probably just require a single sentence.

Our response: In the new version, we have added a paragraph with some equations to explain in a pedagogical way how the spin-orbit coupling gives rise to the anisotropy term.

The referee writes: 2- The statement explaining the degeneracy between the two and one particle states for $ \Gamma=0 $ can be confusing. There are more than one two particle states with $S_z=0$ : $|20>$; $|02>$; $|\uparrow \downarrow>$; $|\downarrow \uparrow>$ (before symmetrization). The authors would help the reader by enumerating or explicitly pointing out what states they refer to and justifying why the others are discarded.

Our response: In the revised version, we write explicitly the five states considered. As stated, they correspond to the ground state for each configuration. Therefore, for a given number of particles (one or two) only the lowest-energy states are considered. For two particles, the singlet states are at energy $ J_H$ or more above the ground state, and only the state $|S=1, S_z =0>$ is relevant for $\Gamma$ tending to zero.

The referee writes: 3- I suspect there is a typo in Eq. 4; the second term is probably $S^{+}S^{-} + H.c.$

Our response: Yes. We have corrected this term. We thank the referee for pointing out this mistake.

The referee writes: 4- In the first sentence of section 5 "the experimental situation of type A" is not defined, what do the authors mean? (type B is later mentioned, but never described). If this requires the addition of a figure, I strongly recommend its addition for clarity.

Our response: The two types of behavior (A and B) were explained in Section 3. In the revised manuscript, we added “(see Section 3 for the explanation of types A and B)”. The detailed atomic structure for these two types of behavior is not known, but different curves of dI/dV were reported, as explained in Section 3.

---

## Round 2 · Referee Report · Anonymous · 2022-11-18

Report

The authors have appropriately answered the questions of the referees, and
I do not have any other comments or questions.
I support the publication of this manuscript in SciPost Physics.

---

## Round 2 · Author Response

We have modified the manuscript taking into account the recommendations of the referees.

---

## Round 2 · List of Changes

- In Section 2, we clarify that the topological transition to a non-Landau phase can be driven not only by increasing the anisotropy D, but also by decreasing the exchange coupling J or the hybridization $\tilde V$

- In Section 4, we add a calculation of the anisotropy taking the simplest part of the spin-orbit coupling to clarify its origin. We also write explicitly the states that are discussed as the relevant ones for small $\Gamma$. Also in this section, we correct the typo in Eq. (6), explain the absence of sharp peaks and dips in Fig. 3, and discuss the asymmetry in the figure.

- In Section 5, we refer to section 3 for the explanation of the two types of experimental behavior observed (A and B).

---

## Editorial Decision

published